# Fine-scale tracking reveals visual field use for predator detection and escape in collective foraging of pigeon flocks

**Mathilde Delacoux[1,2,3]\*, Fumihiro Kano[1,2]\***

[1]Centre for the Advanced Study of Collective Behaviour, University of Konstanz, Konstanz, Germany; [2]Max Planck Institute of Animal Behavior, Konstanz, Germany; [3]International Max Planck Research School for Quantitative Behavior, Ecology and Evolution, Radolfzell, Germany

**Abstract** During collective vigilance, it is commonly assumed that individual animals compromise their feeding time to be vigilant against predators, benefiting the entire group. One notable issue with this assumption concerns the unclear nature of predator 'detection', particularly in terms of vision. It remains uncertain how a vigilant individual utilizes its high-acuity vision (such as the fovea) to detect a predator cue and subsequently guide individual and collective escape responses. Using fine-scale motion-capture technologies, we tracked the head and body orientations of pigeons (hence reconstructed their visual fields and foveal projections) foraging in a flock during simulated predator attacks. Pigeons used their fovea to inspect predator cues. Earlier foveation on a predator cue was linked to preceding behaviors related to vigilance and feeding, such as head-up or down positions, head-scanning, and food-pecking. Moreover, earlier foveation predicted earlier evasion flights at both the individual and collective levels. However, we also found that relatively long delay between their foveation and escape responses in individuals obscured the relationship between these two responses. While our results largely support the existing assumptions about vigilance, they also underscore the importance of considering vision and addressing the disparity between detection and escape responses in future research.

**\*For correspondence:**
mdelacoux@ab.mpg.de (MD);
fkano@ab.mpg.de (FK)

**Competing interest:** The authors declare that no competing interests exist.

## eLife assessment

In this **fundamental** study, the authors use innovative fine-scale motion capture technologies to study visual vigilance with high-acuity vision, to estimate the visual fixation of free-feeding pigeons. The authors present **compelling** evidence for use of the fovea to inspect predator cues, the behavioral state influencing the latency for fovea use, and the use of the fovea decreasing the latency to escape of both the focal individual and other flock members. The work will be of broad interest to behavioral ecologists.

## Introduction

In everyday natural tasks, such as locating food, avoiding predators, and interacting with conspecifics, animals constantly face challenges in deciding when and how to adjust their behaviors to increase their chances of survival (**McFarland, 1977**). One such behavior is the focusing of attention, or 'looking' behavior, which has been relatively understudied in natural tasks due to the technical challenges involved in tracking an animal's gaze during natural activities (but see **Kane et al., 2015**; **Kano et al., 2018**; **Miñano et al., 2023**; **Yorzinski and Platt, 2014**). However, this is relatively well studied within the context of vigilance, a scenario where an animal's survival hinges on effective attentional allocation

**eLife digest** Most animals have to compromise between spending time foraging for food and other resources and keeping careful watch for approaching predators or other threats. Many are thought to address this trade-off by living in a group where they rely on the vigilance of others to free up more time for foraging. If one individual animal detects a threat, they alert the whole group so that every individual can respond. However, it remains unclear how individuals use vision to detect a threat and how they communicate the threat to the rest of the group.

Pigeons are a useful animal model to address this question because they tend to live in groups and their vision is well understood. A pit at the back of their eye called the fovea is responsible for building clear, detailed images of the centre of the field of vision. When pigeons attend to something of interest, they typically direct their gaze by moving their whole head instead of moving their eyes, making head orientation a good proxy for researchers to track where they are looking.

To better understand how pigeons detect potential threats and communicate them to the rest of the flock, Delacoux and Kano used motion capture technology to track the head movements of groups of pigeons. To encourage the pigeons to forage, grain was scattered in the centre of an enclosed room. A plastic sparrowhawk (representing a potential predator) would then emerge and move across the room before disappearing again.

Analysis of the imaging data revealed that pigeons use their fovea to spot predators. Individuals that were looking around before the potential predator emerged directed their fovea towards it more quickly than pigeons that were eating. These pigeons also took flight more quickly, and this likely triggered the rest of the group to follow.

Due to improvements in the tracking technologies, these findings may help scientists understand in finer detail how animals in a group detect and respond to threats and other cues in their environment. Therefore, the experimental approach used by Delacoux and Kano could also be used to investigate how information is passed among groups of other animal species.

and where researchers can observe their scanning behavior even in field conditions (*Cresswell, 1994*; *Evans et al., 2018*; *Inglis and Lazarus, 1981*). Vigilance, especially during foraging, generally involves the compromise between staying alert for threats and searching for food, as well as maintaining a balance between individual and collective vigilance (*Beauchamp, 2015*).

## Common assumptions of vigilance research

Common assumptions about collective vigilance, first coined by *Pulliam, 1973*; *Pulliam et al., 1982*, posit that a vigilant individual, often identified by a 'head-up' posture, is likely to have a higher probability of detecting an approaching predator compared to a feeding individual, typically in a 'head-down' posture. Consequently, the vigilant individual can react and escape more swiftly, thereby reducing its predation risk (also see *Beauchamp, 2015*; *Godin and Smith, 1988*). In a group setting, non-vigilant members can gain benefits from the vigilant individual by following its lead when it exhibits escape behavior (*Pulliam, 1973*). This potential group advantage, known as 'collective detection', might explain that individuals in a larger group tend to be less vigilant (and hence spend more time in foraging), suggesting a critical advantage of grouping (*Pulliam, 1973*; *Pulliam et al., 1982*).

## Empirical challenges

While these assumptions underpin many influential theoretical models in behavioral ecology, they have not gone unchallenged, with numerous empirical studies highlighting potential inconsistencies. First, the trade-off between vigilance and foraging may not be as pronounced as previously assumed. Even though several studies found a relationship between vigilance and escape latency (*Devereux et al., 2006*; *Hilton et al., 1999*; *Lima and Bednekoff, 1999*), various studies have demonstrated that these two behaviors are not necessarily mutually exclusive (*Bednekoff and Lima, 2005*; *Cresswell et al., 2003*; *Devereux et al., 2006*; *Kaby and Lind, 2003*; *Lima and Bednekoff, 1999*). In several species, especially those with a broad visual field and specific retinal structures such as the visual streaks, individuals can simultaneously engage in foraging activities while remaining vigilant (*Fernández-Juricic, 2012*), likely using peripheral vision to detect approaching threats (*Bednekoff*

*and Lima, 2005*; *Cresswell et al., 2003*; *Kaby and Lind, 2003*; *Lima and Bednekoff, 1999*). Relatedly, although vigilance and foraging have been defined in many previous studies, respectively, as head-up and head-down due to the limitation associated with direct observation, several studies have pointed out that vigilance triggered by the appearance of a predator is not necessarily related to the head-up postures but to the pattern of head movements (*Fernández-Juricic, 2012*; *Jones et al., 2007*; *Jones et al., 2009*). Similarly, the patterns of head-up (number of head-up bouts, their length and regularity), rather than the proportion of time being head-up versus head-down, are sometimes better predictors of predator detection (*Beauchamp, 2015*; *Beauchamp and Ruxton, 2016*; *Bednekoff and Lima, 2002*; *Cresswell et al., 2003*; *Hart and Lendrem, 1984*).

Second, it is challenging to empirically differentiate the effect of collective detection from other group size-related effects, such as confusion, dilution, and edge effect. Although there is evidence that non-detectors can benefit from detectors' escape (*Davis, 1975*; *Lima, 1995b*; *Lima and Zollner, 1996*), other studies found evidence for the risk dilution (*Beauchamp and Ruxton, 2008*) and the edge effect (*Inglis and Lazarus, 1981*) in their study systems. It appears that multiple factors influence vigilance across species, including group size, density, and spatial configuration, as well as social contagion (*Elgar, 1989*; *Roberts, 1996*).

Third, non-vigilant animals do not always respond to behavioral cues of other members of the group. The upright-freezing alert posture, often one of the first behavioral indications of predator detection, tends to be rather inconspicuous, and flockmates usually do not seem to respond to it (*Fernández-Juricic et al., 2009*; *Lima, 1995b*). Additionally, birds do not necessarily differentiate between predator-based escapes and non-threat departures from a flock mate, suggesting that simultaneous departures of multiple birds are required to trigger contagious flights (*Cresswell et al., 2000*; *Lima, 1995b*; *Proctor et al., 2001*). As a result, the extent to which group-living animals benefit from collective detection remains open to questions (*Bednekoff and Lima, 1998*).

Finally, the definition of 'detection' is generally uncertain in studies. Previous studies typically used overt escape responses, such as flying, as a measure of detection (using escape as only measure: *Kenward, 1978*; *Lima and Zollner, 1996*; *Quinn and Cresswell, 2005*; or partially using escape as response: *Cresswell et al., 2003*; *Lima, 1995a*, *Lima, 1995b*; *Tisdale and Fernández-Juricic, 2009*; *Whittingham et al., 2004*). However, a bird likely identifies a potential threat before escaping (*Barbosa and Castellanos, 2005*; *Fernández-Juricic, 2012*; *Lima and Dill, 1990*). Thus, subtler behavioral responses like freezing or alertness were also used (e.g., *Fernández-Juricic et al., 2009*; *Kaby and Lind, 2003*; *Lima and Bednekoff, 1999*; *Rogers et al., 2004*). These studies have found a period between these subtle behavioral shifts and overt escape actions (response time delay), which is considered a crucial risk-assessment phase (*Cresswell et al., 2009*). This phase has been found to depend on multiple factors such as species, context (e.g., food availability, surrounding environment: *Fernández-Juricic et al., 2002*), individual characteristics (*Jones et al., 2009*), and flock size (*Boland, 2003*). An issue remains, however, that even these subtle behaviors might not accurately capture the time when an animal detects an approaching threat (*Barbosa and Castellanos, 2005*; *Fernández-Juricic, 2012*; *Lima and Dill, 1990*; *Tätte et al., 2019*), due to the lack of a more direct measure of visual attention.

## Visual sensory ecology of birds

More recent research in visual sensory ecology has emphasized the role of vision in the context of vigilance. Vision is crucial for gathering distant predator cues, and it is thus expected to play a significant role in predator–prey interactions (*Barbosa and Castellanos, 2005*). Specifically, predation is considered one of the primary drivers of the evolution of the visual system in birds, as well as foraging (*Martin, 2017*). Birds' retinal specializations, such as the *area* (a region of the bird's retina with a high density of photoreceptors) or the fovea (a pit-like area in the retina with high concentration of cone cells where visual acuity is highest, and is responsible for sharp, detailed, and color vision see *Bringmann, 2019* for more details), are thought to be crucial for the detection, identification, and tracking of predators (*Fernández-Juricic, 2012*; *Martin, 2017*) as well as risk assessment and response selection (*Cresswell, 1993*; *Cresswell et al., 2009*; *Fuchs et al., 2019*; *Veen et al., 2000*). Experimental studies tracking the foveal projections demonstrated that birds indeed use their foveas to inspect predator cues (*Butler and Fernández-Juricic, 2018*; *Tyrrell et al., 2014*; *Yorzinski and Platt, 2014*). Given the diversity and complexity of birds' visual field configurations, it is necessary to understand

how different types of visual information are acquired based on body posture and head orientation, rather than simply to consider a head-up and head-down dichotomy (*Fernández-Juricic et al., 2004*). Notably, while it is a prevalent notion that vigilant individuals are quicker to detect predators, this fundamental assumption has seldom been directly scrutinized in the literature (*Beauchamp, 2015*).

## Experimental rationales

This study thus leveraged recent fine-scale tracking methods to re-evaluate how a vigilant individual utilizes its high-acuity vision (such as the fovea) to detect a predator cue and subsequently guide individual and collective escape responses in the context of foraging and vigilance. We utilized a state-of-the-art, non-invasive motion-capture technique to monitor the head orientations of pigeons moving freely within a flock (*Kano et al., 2022*; *Nagy et al., 2023*; *Figure 1a, b*). Motion-capture cameras track with high accuracy the three-dimensional (3D) position of markers, which, when attached to the pigeon's head and body, enables to reconstruct the rotations of the head and body in all directions. This approach enabled us to closely analyze the birds' fine-scale looking behaviors, while maintaining their natural foraging, vigilance, and social interactions; before, during, and after simulated predation events (*Figure 1c*).

Pigeons are well suited for our study system because they are relatively well studied for their visual system. In brief, although a relatively high acuity is maintained over the retina (*Hayes et al., 1987*), they have one fovea centrally located in the retina of each eye, with an acuity of 12.6 c/deg (*Hodos et al., 1985*). Their fovea projects laterally at ~75° into the horizon in their visual field. They mainly use their foveas to attend to objects or conspecifics in the distance (>0.5 m; *Kano et al., 2022*; *Nalbach et al., 1990*). Pigeons have another sensitive spot in their retina, the red field, which projects to the ground in their visual field. They mainly use the red fields in a foraging context, to search and peck seed on the ground in a close distance (<0.5 m; *Kano et al., 2022*; *Nalbach et al., 1990*). Pigeons have a binocular overlap in the front covering an angle of approximately 20°, which they mainly use for pecking, perching, or attending to slow moving objects (*Bloch et al., 1984*; *Green et al., 1992*). Their blind area covers an angle of approximately 40° at the back of the head (*Hayes et al., 1987*; *Martin, 1984*). Due to these systematic structures, we assumed that their head movement indicates critical information about vigilance, foraging, and detection. Yet, it should be noted that their eye movement was not tracked in our system, although it is typically confined within a 5 degrees range (*Wohlschläger et al., 1993*). We thus considered this estimation error of the foveation (directing visual focus to the fovea to achieve the clearest vision) in our analysis, as a part of the error margin (see Methods).

In our observations, pigeons foveated on the predator cue typically before the offset of looming stimulus (*Figure 1d*); we considered this foveation as a potential indicator of early detection. Pigeons initiated running and then took flight typically during a predator presentation event (*Figure 1d*, see also *Figure 1—source data 1*). Pigeons responded either directly to the looming stimulus or after the appearance of the model predator, by running away and/or flying. Therefore, our first measure of escape was the presence of an escape response (either running or flying) prior to the looming stimulus offset (i.e., whether the pigeon runs or flies before the looming stimulus disappears). Our second measure of evasion focused on flight responses (predominantly occurring after the predator appeared), given that flight is the most prominent form of escape response in birds and is widely noted in the literature for its contagious nature among flockmates (*Davis, 1975*; *Lima, 1995b*; *Lima and Zollner, 1996*).

## Experimental hypotheses

We subsequently developed a series of hypotheses outlined in *Table 1*. The initial two hypotheses (Hypotheses 1 and 2) aim to examine whether foveation correlates with predator detection. Hypothesis 1 posits that pigeons employ their foveas to evaluate the predator (looming) cue and Hypothesis 2 that the foveation latency is correlated with vigilance-related behaviors of the individual. Hypothesis 2 was further divided into two sub-hypotheses; respectively, assessing if the foveation latency depends on the state of the individual at the onset of the looming cue, and depends on the general behavior of the pigeons before the cue appears. Specifically, Hypothesis 2.1 suggests that the birds' behavioral state observed at the onset of the looming stimulus (e.g., head-up, foraging) predicts the latency to foveate on the cue. Additionally, we assessed spatial positioning—relative to the threat (distance

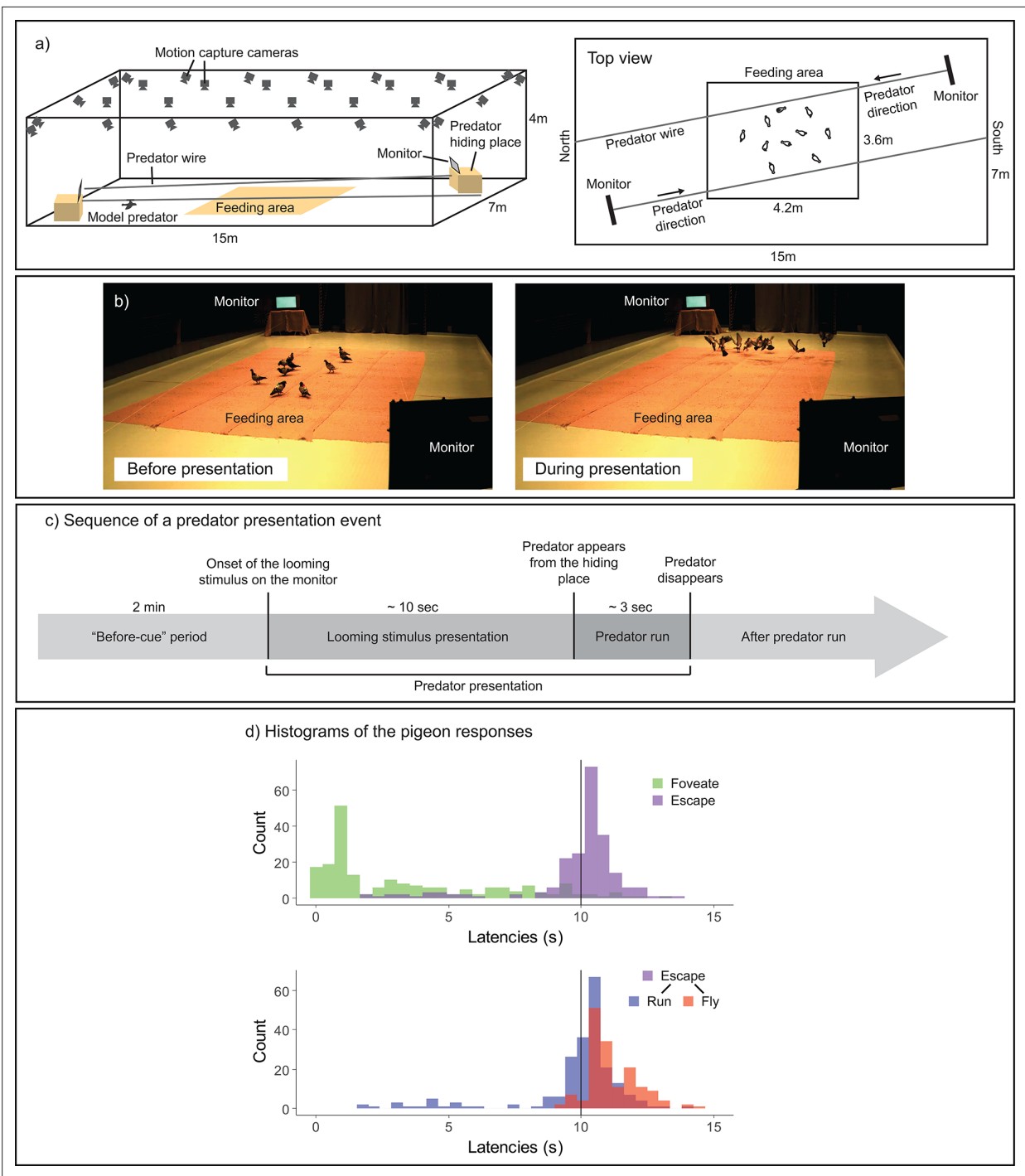

**Figure 1.** Experimental design. (**a**) Experimental setup. In a large-scale motion-capture system (SMART-BARN), the feeding area, the monitors, and the predator running wire were installed (a three-dimensional [3D] view on the left, and a top view on the right). (**b**) Still frames captured from the periods before (top) and during (bottom) the predator presentation, respectively. (**c**) Sequence of an event presenting a model predator. Each predator presentation event started with the presentation of a looming stimulus (predator silhouette), which lasted for approximately 10 s, on either of the two monitors, followed by the model predator running on the wire across the room (lasting for approximately 3 s). The 'before-cue' period is defined as the 2 min directly preceding the onset of the looming stimulus. (**d**) Histograms of the pigeons' responses. Top: latency to foveate (green) and latency to escape—either running or flying (purple); bottom: latency to run (blue) and latency to fly (red) separately. The vertical line shows the looming stimulus offset.

The online version of this article includes the following source data and figure supplement(s) for figure 1:

**Source data 1.** Number of individuals foveating, escaping and flying for each event.

**Figure supplement 1.** Latencies to respond to each predator event.

**Table 1.** A set of hypotheses testing the assumptions about vigilance.

| | Hypothesis | Test | Response | Description of the test |
|---|---|---|---|---|
| 1 | Pigeons foveate on the predator cue (monitor or predator hiding table) during the 10 s presentation period of the looming stimulus. | Linear mixed model (LMM) | Proportion of time foveating on the objects of interest. | Testing the effect of: Object type (the monitor presenting the looming stimulus, the monitor that did not present the looming stimulus, and the nearest neighbor pigeon). |
| 2.1 | Pigeons that are head-up (while not grooming nor courting or courted) foveate on the predator cue earlier than those that are feeding at the onset of the looming stimulus. | LMM | Latency to foveate on the predator cue. | Testing the effects of: Behavior exhibited at the onset of the looming stimulus (head-up/feeding), distance from the monitor presenting the looming stimulus, and distance from the nearest neighbor conspecific. |
| 2.2 | Pigeons' vigilance/feeding-related behaviors during the 2 min 'before-cue' period predict their earlier foveation on the predator cue. | LMM | Latency to foveate on the predator cue. | Testing the effects of: Vigilance/feeding-related behaviors (proportion of time spent head-up, saccade rate, pecking rate, proportion of time foveating on any monitor, proportion of time foveating on any conspecifics). |
| 3 | Pigeons' first foveation on the predator cue predicts their escape. | LMM and generalized linear mixed model (GLMM) | Probability of evasion before the offset of the looming stimulus, latency to fly. | Testing the effect of: Latency to foveate on the predator cue. |
| 4.1 | The first pigeon foveating on the predator (looming) within a flock facilitates the escape behaviors of the other members. | LMM and GLMM | Probability of escape before the offset of the looming stimulus and latency to fly in the other members (excluding the first pigeon). | Testing the effect of: Latency to foveate on the predator cue by the first pigeon. |
| 4.2 | Escape responses within a predator presentation event are more clustered than expected by chance. | Permutation test | Time interval between the successive escape (or flying) responses of pigeons within a given predator presentation event. | Comparing the mean interval observed in the data with the mean intervals expected by chance in a control event created by shuffling the latencies between event ID. |

The online version of this article includes the following source data for table 1:

**Source data 1.** Description of the used variables and responses.

**Source data 2.** R formulas used for the different models.

**Source data 3.** Detailed model results.

**Source data 4.** Repeatability of the used variables.

from the monitor) or to conspecifics (nearest neighbor)—as these factors are known to influence vigilance and escape behaviors (*Beauchamp, 2008*; *Beauchamp and Ruxton, 2008*; *Inglis and Lazarus, 1981*). Hypothesis 2.2 posits that various aspects of vigilance/foraging behaviors observed prior to the presentation of the looming stimulus (e.g., head-up, pecking rate, head-saccade rate, time spent foveating on predator-related objects or conspecifics) correlate with the latency to foveate on the cue. As noted, these related yet distinct facets of visual/foraging behaviors have been identified as differentially related to vigilance and escape (*Cresswell et al., 2003*; *Jones et al., 2007*; *Jones et al., 2009*). From Hypothesis 3, we asked whether earlier foveation on the predator cue predicts quicker escape responses by individual birds. While this is a common assumption in the literature, several factors appear to affect the length of the response time delay (*Cresswell et al., 2009*; *Jones et al., 2009*; *Tätte et al., 2019*). The final hypothesis (Hypotheses 4) focuses on the collective dynamics of detection and escape. Specifically, Hypothesis 4.1 posits that the first pigeon in a flock to foveate on the looming stimulus facilitates the escape behaviors of other members. A related hypothesis, Hypothesis 4.2, posits that the timing of escape initiations is socially contagious, specifically that the escape responses within a predator presentation event are more clustered than would be expected by chance.

## Results

### Overview of observations

We tested 20 pigeons across 6 trials each, with every trial involving a flock of 10 pigeons and consisting of 2 predator presentation events. This design resulted in a total of 240 observations, with each observation representing data from a single event for an individual pigeon. Each model incorporated this number of observations while excluding null cases in which pigeons displayed no response or cases where the system lost track of the pigeons (see Method for the details).

Within a few predator presentation events, our pigeons rapidly decreased their latencies to foveate, escape, and fly, indicating that learning occurred rapidly during these trials (*Figure 1—figure supplement 1*). At the very first predator presentation event, approximately half of the flock members did not foveate on the monitor nor escaped even after the appearance of the model predator (*Figure*

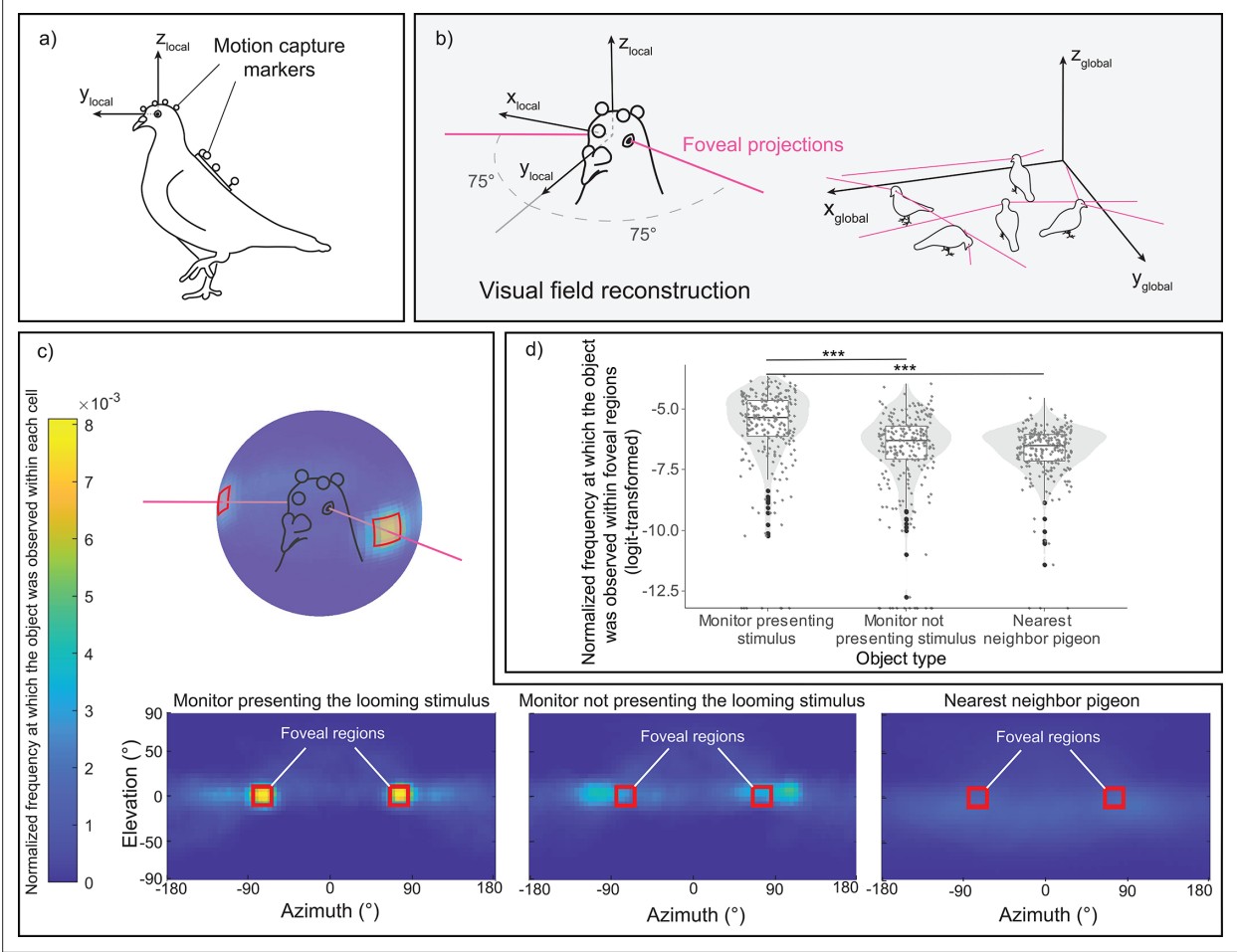

**Figure 2.** Reprojection of the pigeon's visual perspective. (**a**) A pigeon equipped with motion-capture markers on its head and back. (**b**) Reconstruction of estimated foveal projections in the local coordinate system of the head (left) and in the global (motion-capture) coordinate system (right). (**c**) Reprojection of each object of interest—the monitor presenting the looming stimulus, the monitor without the cue acting as a control, and the nearest conspecific—within the visual field of all pigeons across all trials. The red squares denote the designated region of estimated foveal projections in the visual field, defined as 75 ± 10° in azimuth and 0 ± 10° in elevation. The color scale illustrates the normalized frequency (normalized to a unit sum) at which each object of interest was observed within each 5 × 5° cell of the heatmap. (**d**) Normalized frequency (normalized to unit sum then logit-transformed) at which each object of interest was observed within the defined foveal regions on the heatmap. The dots represent single observations (jittered horizontally for visualization), the gray shade represents the violin plot of the distributions, and the boxplot's boxes represent the 0.25 and 0.75 quartiles (with the median represented as a line inside the box) and the whiskers the minimum and maximum values within the lower/upper quartile ±1.5 times the interquartile range. *** represent a significant difference with p < 0.001.

The online version of this article includes the following figure supplement(s) for figure 2:

**Figure supplement 1.** Spherical representation of a pigeon.

1—*source data 1*). From the second predator presentation event, most pigeons made foveation, escape, and flight responses. Our pigeons maintained these responses throughout the trials, although in the last few trials pigeons decreased their flight responses, indicating that habituation was minimal.

## Use of foveas during predator cue inspection (related to Hypothesis 1)

From the markers coordinates, we were able to reconstruct for each pigeon their head orientations simultaneously (*Figure 2a, b*). In the heatmaps (*Figure 2c*), we projected the 3D representations of the three objects of interest—the monitor displaying the looming stimulus, the monitor without the cue serving as a control, and the nearest conspecific (see *Figure 2—figure supplement 1* for the 3D object definitions)—onto the pigeons' local head coordinate system. We sampled these data during the period of the looming stimulus presentation and before any escape response by the pigeon (normalized to unit sum). Visual inspection suggested that pigeons foveated on the monitor displaying the looming stimulus, more so than on the other monitor that was not presenting the looming stimulus or on the nearest conspecific. It should be noted that the two monitors were placed on opposite sides of the room. Consequently, a pigeon located in the center and foveating on one monitor would have the other monitor falling ~105° in azimuth angle on the opposite side of its head. This is likely resulting in the spot next to the foveal region for the monitor not displaying the looming stimulus (see *Figure 2c*).

When testing for the normalized frequency at which each object was observed within the foveal region, we observed a significant effect of the object type ($\chi^2(1) = 108.06$, $p < 0.001$) (*Figure 2d*). Subsequent follow-up analyses revealed that pigeons focused on the monitor displaying the looming stimulus for a longer period compared to the other two objects (vs. monitor not presenting the looming stimulus, $\chi^2(1) = 62.44$, $p < 0.001$; vs. nearest neighbor, $\chi^2(1) = 97.41$, $p < 0.001$).

## Testing the assumptions about vigilance (related to Hypothesis 2)

The analysis predicting the latency to foveate with behavioral state (either head-up or feeding) and spatial factors (the distance from the monitor and from the nearest neighbor) at the onset of the looming cue reveal that it was significantly influenced by the behavioral state ($\chi^2(1) = 13.78$, $p < 0.001$) (*Figure 3a*). However, it was not significantly affected by the other spatial factors, such as the distance from the monitor ($\chi^2(1) = 1.20$, $p = 0.27$) or the distance from the nearest neighbor ($\chi^2(1) = 0.05$, $p = 0.82$).

In addition, the latency to foveate was significantly predicted by most of the feeding-/vigilance-related behaviors of the pigeon during the 'before-cue' period, including the proportion of time spent being head-up ($\chi^2(1) = 15.90$, $p < 0.0001$), the pecking rate ($\chi^2(1) = 14.07$, $p = 0.0002$), the proportion of time foveating on a monitor ($\chi^2(1) = 12.04$, $p = 0.0005$), and the saccade rate ($\chi^2(1) = 9.85$, $p = 0.0017$). However, the proportion of time foveating on any conspecifics did not have a significant effect ($\chi^2(1) = 0.06$, $p = 0.8116$) (*Figure 3b, c*, *Figure 3—figure supplement 1*).

When comparing the Akaike information criterion (AIC) of all five models, the most effective model included the proportion of time spent being head-up (AIC = 609.99). This was followed by models including the pecking rate (AIC = 612.40), the proportion of time foveating on the monitor (AIC = 614.01), and the saccade rate (AIC = 616.00).

In further analyses outlined in *Figure 3—figure supplements 1–3* and *Figure 3—source data 1*f, we examined the changes in these vigilance- and feeding-related behaviors shortly after the predator disappeared (1 min after the predator's disappearance) as compared to the 1-min period preceding the stimulus onset. The results highlighted a significant increase in vigilance and a significant decrease in feeding immediately following the predator's disappearance across all variables (see *Figure 3—figure supplement 3* and *Figure 3—source data 1* for details).

## Detection and escape (related to Hypothesis 3)

The model predicting the probability to escape before the looming stimulus offset indicated that the latency to foveate had no significant effect ($\chi^2(1) = 0.02$, $p = 0.8716$). However, the model predicting the latency to fly revealed a significant effect of the latency to foveate, suggesting that earlier foveation predicted quicker flight responses ($\chi^2(1) = 6.49$, $p = 0.0108$) (*Figure 3d, e*). We observed that the response time delay (the interval between the latency to foveate and the latency to fly) was relatively lengthy and exhibited considerable variation (*Figure 3f*). During this period, pigeons rarely returned

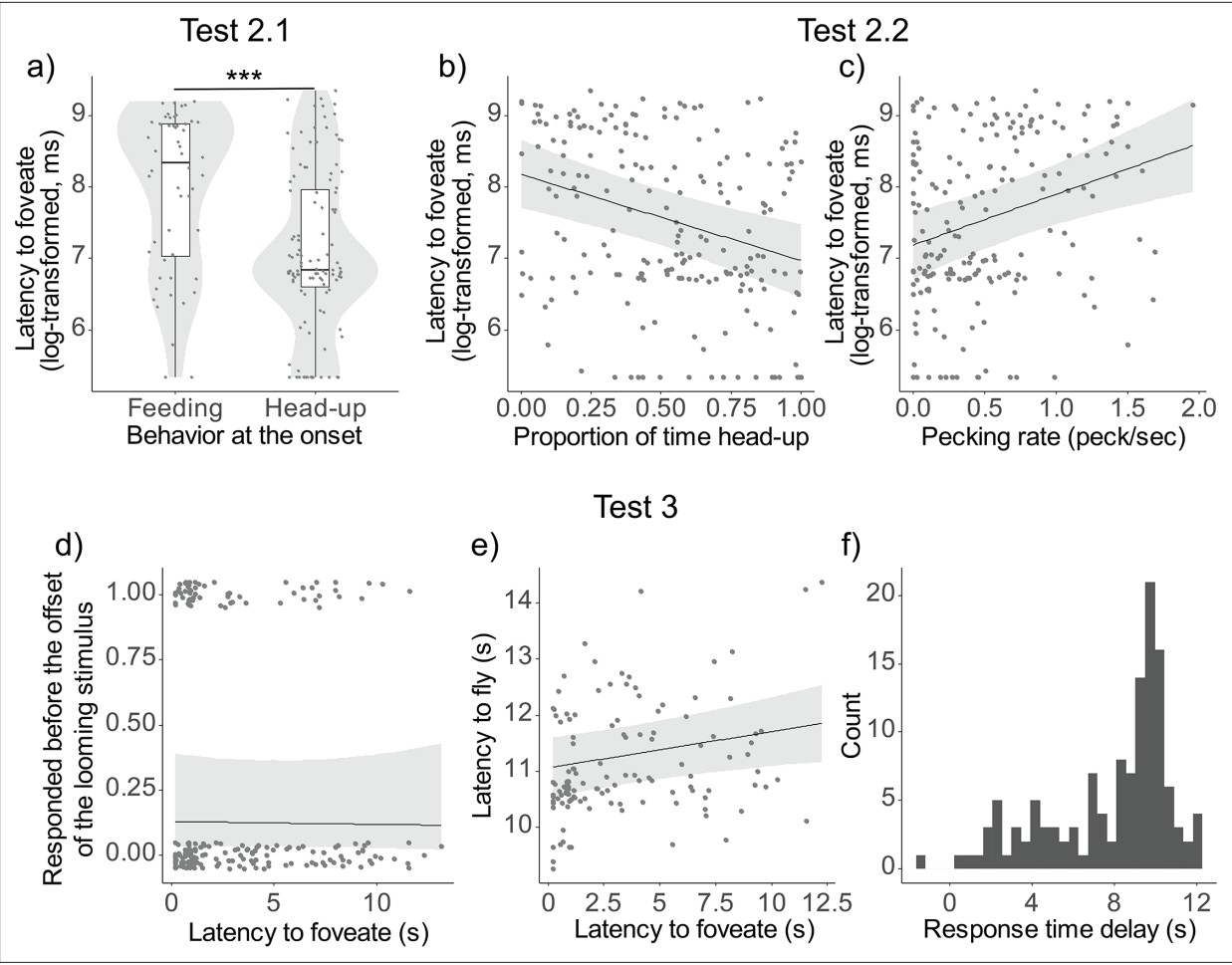

**Figure 3.** Testing assumptions of vigilance, detection and escape. Results of the models from Tests 2 and 3. (**a**) Latency to foveate on the predator cue (log-transformed) as a function of behavioral state (feeding or head-up) at the onset of the looming stimulus. The dots represent single observations (jittered horizontally for visualization), the gray shade represents the violin plot of the distributions, and the boxplot's boxes represent the 0.25 and 0.75 quartiles (with the median represented as a line inside the box) and the whiskers the minimum and maximum values within the lower/upper quartile ±1.5 times the interquartile range. Latency to foveate on the predator cue as a function of the proportion of time spent being head-up during the 'before-cue' period (**b**) and as a function of the pecking rate during the same period (**c**). The graphs for the three other variables can be found in *Figure 3—figure supplement 1*. Probability of escaping before the looming stimulus offset (**d**), and latency to fly (**e**) as a function of the latency to foveate. (**f**) Histogram of the distribution of the response delay (time between the latency to foveate and the latency to fly). For all depicted results, regression lines were determined with other variables held constant, set to their mean values. A comprehensive table detailing the outcomes of the models can be found in *Table 1—source data 3*. *** represent a significant difference with p < 0.001.

The online version of this article includes the following source data and figure supplement(s) for figure 3:

**Source data 1.** Detailed results of the models comparing the behavior of the pigeons before and after an event.

**Figure supplement 1.** Testing assumptions of vigilance: supplementary plots.

**Figure supplement 2.** Behavior of the pigeons before each event.

**Figure supplement 3.** Comparison of the behavior of the pigeons before and after an event.

to the feeding activity, as indicated by the low mean pecking rate (0.10 pecks/s). In *Table 1—source data 4*, we demonstrate that this variation can be partially attributed to individual differences; a within-individual repeatability analysis revealed that one contributing factor to this variation is inter-individual differences ($R$ = 0.128, p = 0.0404).

## Social contagion of escape responses (related to Hypothesis 4)

Earlier foveation of the first pigeon was not significantly related to an earlier escape responses among the other flock members, although there was a trend ($\chi^2(1)$ = 3.66, p = 0.0559). We then examined

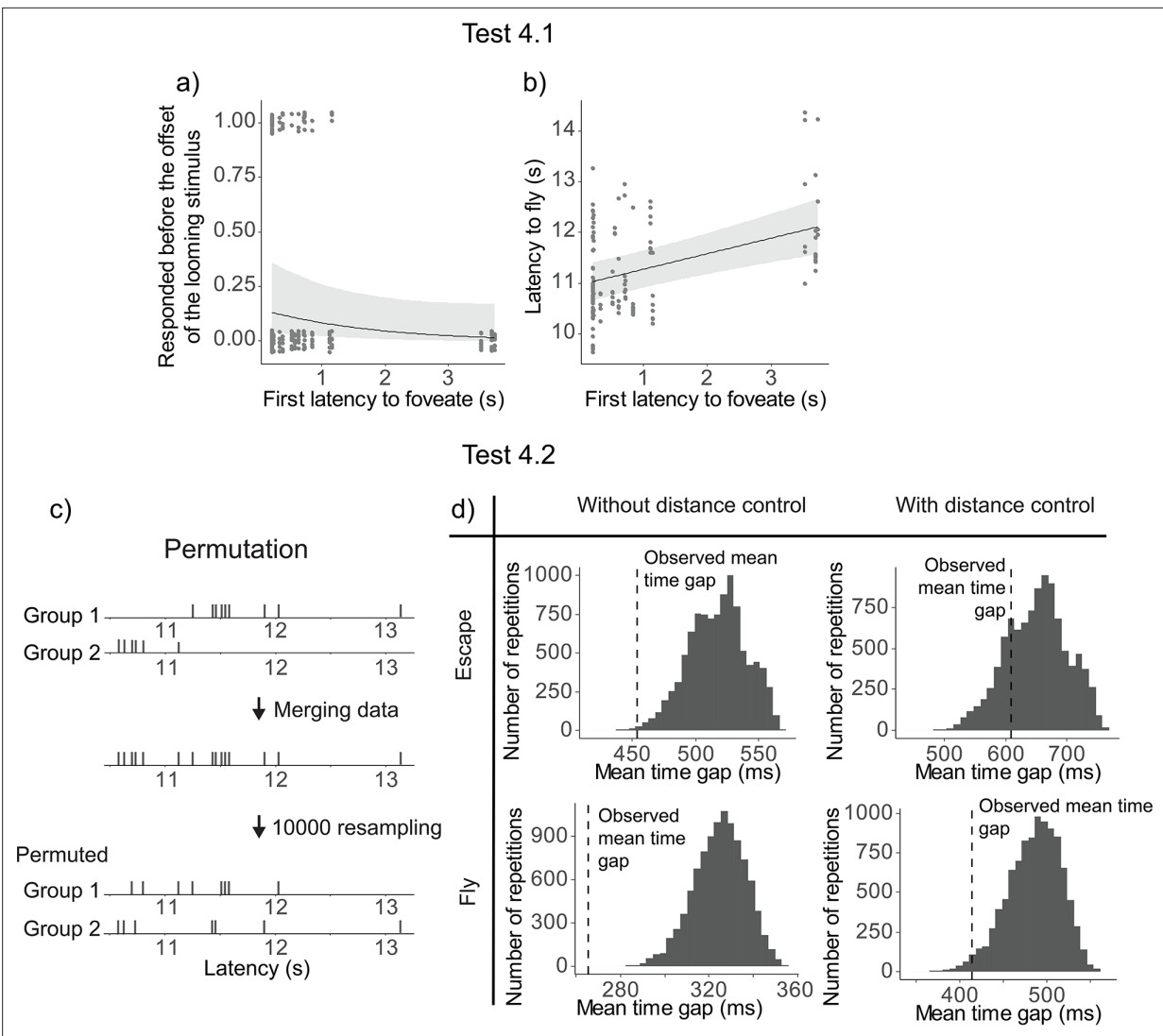

**Figure 4.** Results of Test 4. Latency to foveate of the first individual foveating on the monitor as a function of the other individuals' probability of escape before the cue offset (**a**) and the other individuals' latency to fly (**b**). All the regression lines were calculated with other variables constant and equal to their mean value. A detailed table of the models' results is available in **Table 1—source data 3**. (**c**) An overview of the permutation test procedure. (**d**) Results of the permutation test. Histograms denote the distributions of the average time gap in permutated data. The vertical dashed line indicates the mean time gap from the observed data. The proportion of permutated data less than the observed data (on the left of the dashed line) gives the p-value; escaping (top) and flying (bottom); without (left) or with (right) distance control.

the same model with latency to fly as a continuous response and found that the earlier foveation of the first pigeon significantly predicted earlier flying responses among the other flock members ($\chi^2(1)$ = 17.32, p = 0.0003) (**Figure 4a, b**).

To test Hypothesis 4.2, we conducted a permutation test to check if the time gaps between individual departures were significantly more clustered than expected by chance. In short, we merged departures latencies from two different events, then resampled the latencies to obtain an 'expected mean time gap' between two individual latencies (**Figure 4c**, see Methods for more details). The mean time gaps from the observed data were then compared to those from the 10,000 permutations to determine a p-value. The observed time gaps were significantly shorter than the permuted gaps for both escape (p = 0.00038) and fly responses (p < 0.0001; **Figure 4d**).

Upon inspecting the data, we found a potential confound effect of the distance of the flock from the monitor on the latency to escape or fly, and therefore we ran a second permutation test controlling for the distance (see Methods for more details). Using these controlled datasets, we found that the

mean time gap was not significantly different for the escape latency (p = 0.206), but it was significantly shorter for the fly latency (p = 0.0237) (*Figure 4d*).

## Discussion

This study leveraged fine-scale behavior tracking to examine key assumptions of vigilance research in pigeon flocks during free foraging, particularly focusing on the role of foveation. Using motion-capture technology, we tracked the individuals' visual fields and quantified their behaviors automatically during predator presentation events. Our findings supported the assumptions about vigilance in two aspects: foveation in pigeons is associated with predator inspection, and behaviors related to vigilance and feeding are reliable predictors of earlier foveation in pigeons. Our findings also largely supported the assumptions about collective vigilance in two aspects: earlier foveation is indicative of earlier flight responses at both individual and collective levels, and there is a social contagion effect in their evasion flight responses. However, these results are somewhat complicated by the individuals exhibiting long and variable response time delays—the interval between foveation and escape responses. Moreover, social contagion of evasion was only observed in flight responses following the appearance of the model predator (after most flock members had foveated on the predator cue), not in earlier responses. In summary, while our results largely affirm the prior assumptions about vigilance, we have identified several confounding factors, which will be discussed further below.

### Use of foveas during predator cue inspection (related to Hypothesis 1)

Our pigeons primarily use their foveas to inspect the looming stimulus. This is consistent with previous studies showing that starlings and peacocks use their foveas for predator inspection (*Butler and Fernández-Juricic, 2018*; *Tyrrell et al., 2014*; *Yorzinski and Platt, 2014*). Our research extends these findings to situations where pigeons forage freely on the ground in a flock. It has been previously shown that pigeons use their frontal visual fields for searching and pecking at grain, and their foveas to inspect distantly presented objects (*Kano et al., 2022*). In line with this, our study found that pigeons did not employ their frontal visual fields to inspect the looming stimulus. Additionally, pigeons did not use their foveas to view conspecifics during the stimulus presentation. Collectively, our results indicate that, during the stimulus presentation, pigeons primarily relied on their foveas to attend to the predator cue. This consolidates the idea that the avian fovea is crucial in predator detection and inspection (*Martin, 2017*).

### Testing the assumptions about vigilance (related to Hypothesis 2)

We found that vigilance-related behaviors, including keeping the head-up, engaging in more frequent head movements (saccades), and earlier monitoring of potential threat locations (i.e., monitors), indeed lead to earlier foveation on the predator cue. Conversely, feeding-related behaviors, such as keeping the head-down and a higher pecking rate, delay the foveation on the cue. The spatial configuration of the flock, such as the distance from the predator cue and foveation on conspecifics, did not influence earlier foveation on the predator cue. Further analyses showed that these vigilance-related behaviors significantly increased, and these feeding-related behaviors decreased after the presentation of the predator (see *Figure 3—figure supplement 3*).

These findings indicate that all the measured vigilance- and feeding-related behaviors are relevant to predator detection in our study system. This is in line with several studies (*Devereux et al., 2006*; *Hilton et al., 1999*; *Lima and Bednekoff, 1999*), though not others (*Cresswell et al., 2003*; *Jones et al., 2007*; *Jones et al., 2009*; *Kaby and Lind, 2003*). While prior research has suggested that species with a relatively large visual field can maintain vigilance peripherally (*Fernández-Juricic et al., 2004*), our data propose that behaviors such as head-up and scanning are still advantageous, and head-down while feeding is still costly to predator detection, even in pigeons that possess a relatively large visual field. Furthermore, previous studies have noted that, in blue tits, vigilance- and feeding-related behaviors are linked to detection only when the feeding task is more demanding (*Kaby and Lind, 2003*). This might be partly relevant to our results, as we used a moderately attention-demanding task where pigeons searched for grain in a grain–grit mixture.

Most notably, our fine-scale tracking of pigeon behaviors may have refined our ability to identify predator detection. Previous studies have inferred predator detection by using escape responses

as the sole measure (*Kenward, 1978*; *Lima and Zollner, 1996*; *Quinn and Cresswell, 2005*) or by observing more subtle signs of alertness such as freezing, adopting a straight upright posture, crouching, or ceasing to feed (e.g., *Fernández-Juricic et al., 2009*; *Kaby and Lind, 2003*; *Lima and Bednekoff, 1999*; *Rogers et al., 2004*), or by combining these responses (*Cresswell et al., 2003*; *Lima, 1995a*; *Lima, 1995b*; *Tisdale and Fernández-Juricic, 2009*; *Whittingham et al., 2004*). However, as previously claimed (*Barbosa and Castellanos, 2005*; *Fernández-Juricic, 2012*; *Lima and Dill, 1990*; *Tätte et al., 2019*), predator detection might occur earlier without overt behavioral responses, which can be too subtle to detect under standard observational conditions. In our study, we measured the birds' first foveation on the predator cue while excluding instances of short and early foveation on the cue (see Method). Although making definitive claims is technically challenging without assessing the internal perceptual process, the clear relationships observed between foveation and vigilance- and feeding-related behaviors in our results suggest that foveation is likely a reliable proxy for 'detection' within our study system.

## Detection and escape (related to Hypothesis 3)

In our study, earlier foveation on the predator cue predicted earlier evasion responses from the model predator. However, this relationship was confirmed only when we analyzed the flight responses made after the predator's appearance, not when we included pigeons' running as the escape responses. Moreover, pigeons exhibited relatively long and variable response times between foveation and the flight response. This response time delay was, in part, related to consistent individual differences across trials.

One interpretation of this result is that pigeons, after foveating on the predator cue, assessed its potential risk. The influence of earlier detection on the decision to fly earlier likely stems from increased sensitivity to risk following the appearance of the model predator. The consistent individual differences in response time delay suggest variations in risk sensitivities among individuals. The relatively long and variable response time delay observed is likely inherent to our study design, which involved using a looming stimulus as a warning for the appearance of a model predator. Notably, pigeons rapidly decreased their latency to foveate after just a few presentations (as shown in *Figure 1—figure supplement 1*), indicating increased sensitivity rather than habituation to the stimuli across repeated presentations. The low pecking rate during the response time delay (as detailed in the Results section) suggests that pigeons were alert when viewing the looming stimulus. However, this alertness more likely led them to assess the stimuli rather than to initiate an explicit escape response in most cases. These findings are in line with the previously established idea that detection and escape represent two distinct stages in antipredator responses (*Barbosa and Castellanos, 2005*; *Fernández-Juricic, 2012*; *Lima and Dill, 1990*), and that response time delay is associated with risk assessment (*Cresswell et al., 2009*) and individual differences (*Jones and Godin, 2010*).

## Social contagion of escape responses (related to Hypothesis 4)

When the first pigeon in the flock foveated on the predator cue earlier, the remaining flock members flew earlier. Permutation tests indicated that their evasion flight responses are socially contagious (no clear evidence for earlier escape response involving running).

One interpretation of these results is that the decision to fly by the first pigeon that foveated triggered a following response from the rest of the flock. This social contagion in flight responses has been previously demonstrated in birds, including pigeons (*Davis, 1975*). Notably, our pigeons did not directly foveate on other flock members (as illustrated in *Figure 2*), suggesting that direct foveation on other pigeons is not necessary for the social contagion of flights. This finding is consistent with the notion that birds only need to maintain peripheral visual contact with their flock members to observe and respond to their departure flights (*Lima and Zollner, 1996*). Additionally, considering the findings related to Hypotheses 3 and 4, our results indicate a nuanced form of collective detection: while earlier detectors may trigger earlier flight responses in the flock via social contagion, non-detectors do not necessarily gain an advantage from these detectors, as most flock members likely already detected the predator cue.

## Limitation

A potential limitation of our study is its limited generalizability. We focused on pigeons, which are middle-sized birds, and their escape responses may differ from those of smaller birds commonly studied in vigilance research; pigeons might, for instance, be more hesitant to take flight due to the higher energy costs related to their weight. Furthermore, while our system precisely tracked freely foraging birds, some elements of our experimental setup were artificial. The looming stimulus and the model predator were presented in a somewhat disjoint manner. Pigeons had to rapidly learn the association between these two stimuli over several trials, potentially increasing their reluctance to fly. This learning process and the disjoint stimulus presentation might have contributed to the long and variable response time delays observed in our study. While these delays provided a unique insight into collective escape, particularly in cases where early detection does not immediately result in early escape, it is possible that this scenario might be more likely with a more naturalistic predator stimulus. Future research could address these limitations by conducting similar studies in the field, outside of the motion-capture system, perhaps using the emerging technology of markerless 3D posture tracking in birds (*Waldmann et al., 2023*; *Waldmann et al., 2022*).

## Conclusion

Our research largely supports the common assumptions in vigilance studies. Key among our findings is the relationship between foveation and predator inspection, along with vigilance- and feeding-related behaviors. This suggests that foveation can be a useful proxy for detection in bird vigilance studies. Additionally, we observed that earlier foveation led to earlier flight responses in the flock, facilitated by social contagion. Interestingly, this contagion occurred even though most flock members likely detected the predator cues, as suggested by the long and variable response time delay, which are likely associated with risk assessment. Therefore, our results imply that collective escape does not always involve non-detectors benefiting from detectors. Our study highlights the importance of considering the vision as well as the disparity between detection and escape responses in future vigilance research.

## Methods

### Subjects

Twenty pigeons (*Columba livia*, 13 females and 7 males) participated in this study (492 ± 41 g; mean ± standard deviation [SD]). All pigeons originated from a breeder and were juveniles of the same age (1 year old). They were housed together in an aviary ($2w \times 2d \times 2h$ m; $w$—width, $d$—depth, $h$—height) with perching and nesting structures. The pigeons were fed grains once a day. On experimental days, they were fed only after the experiments was completed; this ensures 24 hr no feeding at the time of the experiment, although we did not control the amount of food over the course of the experimental periods. Water and grit were available in the aviary *ad libidum*.

### Experimental setup

The experiment took place in SMART-BARN, the state-of-the-art animal tracking system build at the Max Planck Institute of Animal Behavior (*Figure 1a*; *Nagy et al., 2023*). We used the motion-capture feature of SMART-BARN, which was equipped with 32 motion-capture cameras in an area of $15w \times 7d \times 4h$ m (12 Vero v2.2 and 20 Vantage 5, VICON). At two opposite corners of the room, we placed tables covered with fabric ('predator hiding place'; $105w \times 75d \times 75h$ cm), which hide from the pigeons' view a model predator (plastic sparrow hawk; wingspan 60 cm and beak-tail length 35 cm). At the center of the room, we placed jute fabric ($4.2l \times 3.6w$ m; $l$—length) where we scatted food to encourage the pigeons to stay there during the experiments (feeding area). The model predator ran on a thin wire across the room via a motored pulley mechanism (Wiral LITE kit, Wiral Technologies AS) until it disappeared into another hiding place set at the end of the room. On top of each predator hiding table, we installed a monitor ($61.5w \times 37h$ cm, WQHD ($2560 \times 1440$), 144 Hz, G-MASTER GB2760QSU, Iiyama), which displayed a looming stimulus before the model predator appears from the hiding place. We specifically chose a monitor with high temporal resolution to match the pigeon's Critical Flicker Fusion Frequency (threshold at which a flickering light is perceived by the eye as steady) that reaches up to 143 Hz (*Dodt and Wirth, 1954*).

This looming stimulus was a silhouette of a predator mimicking the approach of a predator toward the pigeon flock. The looming stimulus lasted for 10 s, starting from a still small silhouette of a predator ($20w \times 4h$ cm) in the first 5 s and then expanding until the silhouette covers the screen entirely in the last 5 s. A looming stimulus was chosen because it is generally perceived as threatening across species (*Evans et al., 2019*), and is subtle enough so that pigeons require effort to detect it in our experimental setup. The looming stimulus was played back on a laptop PC (ThinkPad P17 Gen 2, Lenovo) connected to both monitors, and its onset was manually triggered by the experimenter. This onset time was recorded in the motion-capture system via analog electric signal, specifically by the experimenter pressing a button at the onset of the looming stimulus on a custom Arduino device (Arduino UNO R3, Arduino) connected to the motion-capture system. The same Arduino device also indicated the onset of the model predator to the experimenter via a small LED flash.

## Experimental procedures

### General experimental design

Each pigeon underwent maximum one trial per day and a total of six trials. A flock of 10 pigeons participated in each trial, yielding 2 separate flocks in each trial, and a total of 12 trials in the whole experiment. Each pigeon was pseudo-randomly assigned into either flock in each trial, in such a way that each pigeon was paired within a group with any other pigeon at least once across all six trials. Each trial composed of 2 predator presentation events, and each event presents a model predator from either the hiding place ('North' or 'South' side), yielding 24 events in the whole experiments. The order of presenting the North and South side of hiding places was counterbalanced across groups within each trial.

### Preparation for the experiment

Before the experiment, the pigeons were transported from the aviary using a pigeon carrier ($80w \times 40d \times 24h$ cm). They were then equipped with four motion-capture markers (6.4 mm diameter, OptiTrack) glued on the head feathers and four motion-capture markers (9 mm diameter) on a small solid styrofoam plate ($7l \times 3.5w$ cm); this small plate was attached to a backpack worn by each pigeon with Velcro. The experimenter then held the pigeon's head briefly (less than a minute) in a custom 3D frame equipped with four web cameras and a triangular scale ($26w \times 29d \times 27h$ cm) and then filmed the head with web cameras (C270 HD webcam, Logitech) synchronized by a commercial software (MultiCam Capture, Pinnacle Studio) in order to reconstruct the eyes and beak positions relative to the markers post hoc (see below for the reconstruction methods). Additionally, the motion-capture system was calibrated just before the experiment starts with an Active Wand (VICON) until all cameras have recorded 2000 calibration frames.

### Trial design

At the start of each trial, all 10 pigeons were released in the motion-capture room. After 2 min of acclimatization period, an experimenter scattered a grain–grit mix evenly in the feeding area (*Figure 1a, b*). This grain–grit mix comprised of 200 g of seeds with 400 g of grit to make the foraging task moderately challenging. The experimenter then hid behind a curtain, outside the motion-capture room. After the experimenter visually confirmed that minimally half of the pigeons (≥5 individuals) started feeding (pecking grain; this always happened within a few seconds after the experimenter started scattering the food), a free feeding period started. The duration of this period was randomized between 2 and 5 min across trials to reduce predictability of the predator event. The last 2 min of this feeding period, referred as the 'before-cue' period, were used to extract pre-stimulus behavioral measures. At the end of the feeding period, the experimenter triggered the looming stimulus. After the offset of the looming stimulus (mean 10.83 ± SD 0.43 s after the onset), the experimenter ran the model predator on the wire, and the predator took approximately 3 s before it disappeared from the room (mean 2.83 ± SD 1.18 s; *Figure 1c*). After half of the pigeons resume feeding (within 1–4 min after the predator disappears), and another free feeding period of 2–5 min, a second predator presentation event occurred following the same procedures, except that the looming stimulus and the model predator appeared from the other monitor and hiding place. Each trial lasted for approximately 30 min. After the experiment, the motion-capture markers were detached from each pigeon before being brought back to the aviary.

## Data analysis

### Reconstruction of the pigeons' head

A custom pipeline (see Data availability section) was used to reconstruct the relative 3D position of the center of the eyes, the beak tip and the center of the markers from the four still images of the pigeon's head (see above). These key points were manually labeled by an experimenter on each picture (an updated algorithm is now available for automatic labeling using YOLO, also included in the deposit). A custom MATLAB program based on structure-from-motion reconstructed the 3D coordinates of all the keypoints with a less than 2 pixels mean reprojection error. These procedures were confirmed to yield accurate reconstruction of head orientations, less than a degree of rotational errors (*Itahara and Kano, 2022*).

### Processing of the motion-capture data

The motion-capture data were exported as csv files from the motion-capture software (Nexus version 2.14, VICON). The csv files were then imported and processed in the custom codes written in MATLAB (provide in *Kano et al., 2022*, also provided in our deposit). The motion-capture data consisted of a time series of 3D coordinates of the markers attached to the head and back of the pigeons (*Figure 2a*). From the reconstructed 3D positions of eye centers and beak tip, the local coordinate system of the pigeon's head (the location of the objects and conspecifics from the pigeon's perspective) was defined (*Figure 2b*). In this local coordinate system, the horizon (the elevation of the local $X$ and $Y$ axes in the global coordinate system) was 30° above from the principal axis of the beak, and the $X$, $Y$, and $Z$ axes pointed to the right, front, and top of the pigeon's head, respectively. It should be noted that this angle of 30° was determined based on the typical standing postures of pigeons, as identified in *Kano et al., 2022*. While this previous study determined the angle on a trial-by-trial basis, our study employed a fixed angle following previous recommendations, and also due to the small variation of this angle across trials and individuals.

The data were filtered using the custom pipelines described in Figure S8 of *Kano et al., 2022*. Briefly, the raw motion-capture data were gap-filled in NEXUS, then smoothed using custom MATLAB codes. The translational and rotational movements of the reconstructed local coordinate system were further smoothed, and any improbable movements were removed using the same pipelines. As a result, the loss of head local coordinate system data amounted to 10.07 ± 11.79% (mean ± SD) per individual in each trial, and the loss of back-marker centroids was 2.53 ± 0.98% (mean ± SD). Although the loss of head data was nontrivial, visual confirmation showed that most loss occurred when pigeons were self-grooming (thus self-occluding their head markers from the motion-capture cameras); that is, when they likely were not attending to any object of interest. Head-saccades were also removed from all foveation data due to the possibility that visual processing is inhibited during saccades in birds (*Brooks and Holden, 1973*). It should be noted that one deviation from the previous study in terms of filtering parameters was that the data smoothing was performed at 60 Hz, rather than 30 Hz, following previous recommendations.

### Reconstruction of gaze vectors

We then reconstructed the gaze (or 'foveal') vectors of each pigeon based on the known projected angles of the foveas, 75° in azimuth and 0° in elevation in the head local coordinate system (*Nalbach et al., 1990*). Pigeons use these gaze vectors primarily when attending to an object/conspecific in the middle to far distance (roughly >50 cm from the head center; *Kano et al., 2022*). Although pigeons have another sensitive region of retina, known as red field (*Wortel et al., 1984*), this region was not examined in this study because pigeons primarily use this retinal region to attend to objects in the close distance (roughly <50 cm), such as pecking and searching for grain (*Kano et al., 2022*).

### Behavioral classification

'Foveation' was defined when an object of interest (such as the monitor or another pigeon) fell within ±10° of the gaze vectors. This margin was estimated to accommodate well the eye movement, which was typically within 5° in pigeons (*Wohlschläger et al., 1993*). Head-saccades were defined as any head movement larger than 5°, that lasted for at least 50 ms and faster than 60°/s, and fixations were defined as inter-saccadic intervals, based on the previous study (*Kano et al., 2022*). Our objects of

**Table 2.** Description of the behavior classification definitions.

Terms definitions: state = behavior that lasts in time; event = punctual behavior. Body–head vector = the vector originating from the body center and projecting to the head centroid. Head direction vector = the vector projecting to the front of the head and toward the horizon when the head is still (corresponds to the $y$ axis of the local coordinate system, projecting 30° above the beak–head center axis). Pitch = head rotation movement corresponding to 'nodding' (pitch = 0° corresponds to the head direction vector pointing toward horizon, pitch <0° corresponds to the head direction vector pointing down, pitch >0° corresponds to the head direction vector pointing up). Roll = head rotation movement where the head is 'tilted' (roll = 0° corresponds to a straight head and roll ≠ 0° corresponds to a head tilted to the right or to the left).

| Behavior | Category | Definition |
|---|---|---|
| Head-down | State | The centroid of the head (middle point between the eyes) is lower than the centroid of the body (6 cm below the backpack markers center point) along the $z$ axis of the global coordinate system (=vertically). |
| Head-up | State | The pigeon is not head-down and not exhibiting another behavior (feeding, grooming, or courting). |
| Pecking | Event | The beak tip is closer than 4 cm from the ground and is not oriented toward the body (pitch >−100° from horizon). The pecking event is defined as the frame where the beak tip is the closest from the ground. |
| Feeding | State | Any time between two pecks spaced no more than 6 s apart. |
| Grooming | State | Preening the back feathers: This occurs when the head is pointing backward (defined as when the head direction vector and the body–head vector form an angle larger than 60°) and the beak tip is close to the body center (less than 10 cm) OR when the head is pointing backward (defined as when the head direction vector and body–head vector form an angle larger than 40°) and the beak tip is close to the body centroid (less than 10 cm), and the head centroid is above the body centroid.<br>Preening the breast feathers: This behavior occurs when the head is strongly pointing down (pitch <−100°) OR when the head is pointing down (pitch <−80°) and the head direction vector and the body–head vector form an angle larger than 60°.<br>Scratching the head with the paw: This behavior is exhibited when the roll of the head is large (>50°) and the head is low (less than 3 cm above the body center). |
| Courting | State | The pigeon is bowing its head, which is a typical head movement in pigeon courtship, at a rate of at least two bows within an 8-s period, while being within a proximity of less than 60 cm and oriented toward another pigeon within an angle of less than 120°. We filtered out any instances shorter than 1 s for robust detection. |
| Running away from the monitor | State | When the body centroid of the pigeon moves at a speed faster than 0.6 m/s, and the pigeon is not classified as courting or being courted, while also increasing its distance from the monitor at a speed of 0.2 m/s, we filter out any events shorter than 70 ms for robust detection. |
| Flying | State | When the body centroid of the pigeon moves at a speed greater than 2 m/s, we filter out any events shorter than 70 ms for robust detection. |

The online version of this article includes the following source data for table 2:

**Source data 1.** Histogram of the pigeon's speed over the course of the trials (vertical line representing the running speed threshold).

**Source data 2.** Inter-rater reliablity scores for the automated classification of behaviors.

interest included the monitors, the tables that hid the predator, and conspecifics. These objects were defined as sphere encompassing them (see *Figure 2—figure supplement 1*). In our observations, pigeons foveated on the predator cue (monitor or predator hiding table) typically before its offset (*Figure 1d*); in a few instances (9 out of 120 observations), pigeons foveated on the model predator after the looming stimulus had disappeared, but these cases were excluded from our analysis. To ensure that the first foveation of a pigeon on the predator cue was not a result of random gaze crossing, we excluded instances of foveation that were shorter than 300 ms, based on the typical intersaccadic interval for a pigeon, which typically ranges from 300 to 400 ms (*Kano et al., 2022*). Moreover, to remove cases where the gaze vector was on the predator cue at the onset of the looming stimulus, we excluded from the analysis the first 200 ms of looming stimulus period based on typical reaction times of a pigeon, which is usually longer than 200 ms (*Blough, 1977*).

In addition to the foveation, the pigeons' behavior (e.g., head-up, grooming, feeding, running, and flying) was classified automatically based on simple thresholds using the 3D postural data (*Table 2*). The reliability of the automated classification was verified by two human raters through manual coding. This process was conducted on a dataset different from the one used in this study. The raters assessed the presence or absence of specific behaviors (or the number of pecks) in 1 s sample segments (for coding peck counts, 3 s segments were used to increase variability in the number of pecks). We coded 50 segments for each behavior, with the exception of flying (for which only 30 segments were coded

due to its scarcity in the dataset) and pecking (for which 60 segments were coded to enhance variation in peck counts). Behaviors such as head-down, head-up, and feeding were not rated, as they were determined based on a simple threshold definition and/or depended on rated behaviors. Inter-rater reliability was calculated using R (*R Development Core Team, 2022*) and the 'irr' package (*Gamer, 2019*). The pecking count IRR was computed using intraclass correlation coefficient (ICC) for count data, while grooming, courting, and flying were analyzed using Cohen's kappa for binary, presence–absence data.

The head-up/down state was defined based on the relative position of the head and body. The head-up state was further refined to exclude periods of feeding, grooming, and courting, to avoid moments when pigeons were likely distracted by these activities. To verify the detection of pecks, two human raters counted the number of pecks in short data segments. The ICC confirmed a very high agreement between their coding and automated detection (ICC_min = 0.98; see *Table 2—source data 2*). Feeding was then defined based on consecutive pecking instances. Grooming and courting were identified by a combination of several postural states and validated by the same two human raters (presence or absence of the behavior in short data segments; Cohen's kappa_min = 0.8; see *Table 2—source data 2*). Running was defined based on a speed threshold, determined by inspecting locomotive speed histograms (*Table 1—source data 1*). This definition was limited to pigeons moving away from the predator hiding place and not engaged in courting activities, to identify 'running away from the monitor' as an evasion response. Flying was observable when a pigeon took off but was specifically defined by the locomotive speed exceeding 2 m/s, a threshold unattainable by running alone. This threshold was further validated by the human raters' coding (presence or absence of the behavior in short segments; Cohen's kappa_min = 0.93; see *Table 2—source data 2*). All rated behaviors achieved Cohen's kappa or ICC scores above 0.8 ('almost perfect'; *Landis and Koch, 1977*), indicating a strong agreement between human raters and the automated behavior coding.

## Statistical analysis

All the statistical analyses were performed in R (*R Development Core Team, 2022*). We mainly relied on linear mixed models or generalized linear mixed models (LMMs or GLMMs; lmer or glmer function from the lme4 package; *Bates et al., 2015*) unless otherwise mentioned, and the significance of the predictors was tested using likelihood ratio test. We verified the model assumptions by checking the distribution of the residuals in diagnostic plots (histogram of the residuals, *qq*-plot and plot of the residuals against fitted values). We transformed the response variable (logit- or log-transformation) when it improved the normality of the residual distribution. To test for collinearity, we also checked the variance inflation factor of the predictors. In all models, the continuous predictors were normalized (*z*-transformed). For all models, we also included several control variables to ensure they did not confound with our test predictors: the event number within a trial (1 or 2), the predator side ('North' or 'South'), and the sex of the subject, as well as the pigeon ID and the trial ID as random effects (see *Table 1—source data 2* for the used R formulas). We report the effects of test variables in the Result section and report the effects of all test and control variables in *Table 1—source data 3*. Detailed descriptions about the responses and test variables can be found in *Table 1—source data 1*.

Specifically, to test Hypothesis 1, we quantified the frequency (normalized to a unit sum) at which each object of interest was observed within the defined foveal regions on the heatmap (illustrated by red rectangles; corresponds to the fovea location, adjusted by ±10 degrees in both elevation and azimuth). This quantification was conducted for each pigeon at every predator presentation event. After logit-transforming this response to improve the normality of residuals, we conducted an LMM predicting the normalized frequency at which an object was observed in the foveal region with the object of interest as a within-subject categorical test variable, in addition to control variables and random effects (see *Table 1—source data 3*).

To test Hypothesis 2.1, we conducted an LMM using the latency to foveate (log-transformed to improve the normality of residuals) as the response variable. The pigeons' behavioral state (either head-up or feeding) was included as a categorical test variable, while the distance from the monitor and the distance from the nearest neighbor were included as continuous test variables (in addition to the control variables and random effects; *Table 1—source data 3*).

To test Hypothesis 2.2, we ran five LMMs, each of which include the latency to foveate as a response variable and any of the five related behaviors (the proportion of time spent head-up, the saccade

rate, the pecking rate, the proportion of time foveating on the monitor, and the proportion of time foveating on any conspecifics) as a continuous test variable (in addition to the control variables and the random effects). We further tested the relative predictive power of the different test variables by comparing the resulting models' efficiency using AIC scores.

To test Hypothesis 3, we conducted a binomial GLMM analysis using the probability of escape before the looming stimulus offset as a binary response variable and the latency to foveate as a continuous test variable (in addition to control variables and random effects). Subsequently, we analyzed an LMM with the latency to fly as a continuous response variable, with the same test and control variables.

To test Hypothesis 4.1, we included into a GLMM the probability of escaping before the looming stimulus offset as a binary response and the first pigeon's latency to foveate as a test variable (in addition to control variables and random effects). To test the effect on the flying responses, we ran an LMM with the latency to fly of the other pigeons as continuous response variable, with the same test and control variables.

To test Hypothesis 4.2, we conducted a permutation test based on time gaps between the latencies of individual pigeons to either escape or fly within a given event. This time gap was defined as the latency of the focal individual minus the latency of the immediately preceding individual. For the permuted data, we sampled one event from two different flocks of pigeons. For each sample, the model predator was presented to both flocks from the same side (North or South) during the same trial. We matched the trial and predator presentation side because, among the three control factors (trial, predator presentation side, and event), these two significantly influenced individual latencies as determined by an LMM (the event ID was not a significant factor in this confirmatory test). We combined the latency data from both events, randomly redistributed these data between the two events 10,000 times, and then recalculated the time gaps each time (*Figure 4c*). We then calculated whether the observed mean time gap was significantly smaller than the expected mean time gap from the permutation by calculating the proportion of permutation with an average time gap smaller than the observed one (corresponds to the p-value). We also considered the possibility that the individuals' distance from the monitor could confound the observed effect. Specifically, if the two pigeon flocks occupied different locations across the two events, and the distance from the monitor influenced the individuals' latency to escape or fly, the results might reflect spatial rather than social influences. Upon inspecting the data, this effect seemed plausible. To account for this potential confound, we determined the overlapping areas by calculating the range between the minimum and maximum distances of each flock and subsequently analyzed only the individuals within this overlapping area.

## Acknowledgements

We thank the members of the Max Planck Institute of Animal Behavior (MPI-AB) as well as the Centre for the Advanced Study of Collective Behaviour (CASCB) for their valuable supports in conducting this study. Special thanks are extended to Drs. Mate Nagy, Dora Biro, Oliver Deussen, and Iain D Couzin, along with the members of MPI-AB and CASCB, for their insightful comments on our study. We also thank Drs. Inge Müller and Daniel Zuniga, as well as the caretakers, for their dedicated hosting and care of the pigeons. Finally, we thank Mathias Günther and Alex Chan for technical support. This study was financially supported by MPI-AB, the DFG Cluster of Excellence 2117 CASCB (ID: 422037984), and the CASCB BigChunk projects (ID: L21-07).

## Additional information

### Funding

| Funder | Grant reference number | Author |
| --- | --- | --- |
| Deutsche Forschungsgemeinschaft | 422037984 (L21-07) | Fumihiro Kano |

| Funder | Grant reference number | Author |
|--------|----------------------|--------|

The funders had no role in study design, data collection, and interpretation, or the decision to submit the work for publication. Open access funding provided by Max Planck Society.

## Author contributions

Mathilde Delacoux, Conceptualization, Data curation, Software, Formal analysis, Investigation, Visualization, Methodology, Writing – original draft, Writing – review and editing; Fumihiro Kano, Conceptualization, Resources, Data curation, Software, Formal analysis, Supervision, Funding acquisition, Validation, Investigation, Visualization, Methodology, Writing – original draft, Project administration, Writing – review and editing

## Author ORCIDs

Mathilde Delacoux ⓘ https://orcid.org/0000-0001-5068-5894
Fumihiro Kano ⓘ https://orcid.org/0000-0003-4534-6630

## Ethics

All the experiments using animals in this study were performed under the license 35-9185.81/G-19/107 awarded by the Regierungspräsidium Freiburg, Abteilung Landwirtschaft, Ländlicher Raum, Veterinär- und Lebensmittelwesen, animal ethics authorities of Baden-Wýrttemberg, Germany. Handling was reduced to a minimum to avoid stress. No animal was hurt or killed during the predator presentation tests, and they were transferred back to their home aviary directly after the tests. The pigeons were provided with perches and nesting structures in their aviary, and their health states were checked on a daily basis. After all experiments were performed, all the individuals were given to a breeder to continue living as breeding pigeons.

Reviewer #1 (Public Review): https://doi.org/10.7554/eLife.95549.3.sa1
Author response https://doi.org/10.7554/eLife.95549.3.sa2

# Additional files

## Supplementary files

• MDAR checklist

## Data availability

A OSF repository containing the code used for the processing and analysis can be found on the OSF website (*Delacoux and Kano, 2023*). Sample data are also provided for the user (the whole raw dataset could not be included for size reason). The repository contains the code for the automatic labeling of the head calibration of the pigeons, the Matlab pipeline used for the motion-capture data processing and filtering, and the statistical analyses in R (along with the tables containing the processed data).

The following dataset was generated:

| Author(s) | Year | Dataset title | Dataset URL | Database and Identifier |
|-----------|------|---------------|-------------|------------------------|
| Delacoux M, Kano F | 2023 | Fine-scale tracking reveals visual field use for predator detection and escape in collective foraging of pigeon flocks | https://osf.io/d682s/ | Open Science Framework, 10.17605/OSF.IO/D682S |

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
