## [Editor Report · eLife assessment]

In this **fundamental** study, the authors use innovative fine-scale motion capture technologies to study visual vigilance with high-acuity vision, to estimate the visual fixation of free-feeding pigeons. The authors present **compelling** evidence for use of the fovea to inspect predator cues, the behavioral state influencing the latency for fovea use, and the use of the fovea decreasing the latency to escape of both the focal individual and other flock members. The work will be of broad interest to behavioral ecologists.

---

## [Referee Report · Reviewer #1 (Public Review)]

The authors used an innovative technic to study the visual vigilance based on high-acuity vision, the fovea. Combining motion-capture features and visual space around the head, the authors were able to estimate the visual fixation of free-feeding pigeon at any moment. Simulating predator attacks on screens, they showed that (1) pigeons used their fovea to inspect predators cues, (2) the behavioural state (feeding or head-up) influenced the latency to use the fovea and (3) the use of the fovea decrease the latency to escape of both the individual that foveate the predators cues but also the other flock members.

The paper is very interesting, and combines innovative technic well adapted to study the importance of high-acuity vision for spotting a predator, but also of improving the behavioural response (escaping). The results are strong and the models used are well-adapted. This paper is a major contribution to our understanding of the use of visual adaptation in a foraging context when at risk. This is also a major contribution to the understanding of individual interaction in a flock.

---

## [Author Response]

The following is the authors’ response to the original reviews.

**eLife assessment**
In this fundamental study, the authors use innovative fine-scale motion capture technologies to study visual vigilance with high-acuity vision, to estimate the visual fixation of free-feeding pigeons. The authors present convincing evidence for use of the fovea to inspect predator cues, the behavioral state influencing the latency for fovea use, and the use of the fovea decreasing the latency to escape of both the focal individual and other flock members. The work will be of broad interest to behavioral ecologists.

We thank the editor for his interest and feedback on the manuscript. We hereafter addressed the comments of the reviewer.

**Reviewer #1 (Public Review):**
Summary:The authors were using an innovative technic to study the visual vigilance based on high-acuity vision, the fovea. Combining motion-capture features and visual space around the head, the authors were able to estimate the visual fixation of free-feeding pigeon at any moment. Simulating predator attacks on screens, they showed that (1) pigeons used their fovea to inspect predators cues, (2) the behavioural state (feeding or head-up) influenced the latency to use the fovea and (3) the use of the fovea decrease the latency to escape of both the individual that foveate the predators cues but also the other flock members.Strengths:The paper is very interesting, and combines innovative technic well adapted to study the importance of high-acuity vision for spotting a predator, but also of improving the behavioural response (escaping). The results are strong and the models used are well-adapted. This paper is a major contribution to our understanding of the use of visual adaptation in a foraging context when at risk. This is also a major contribution to the understanding of individual interaction in a flock.Weaknesses:I have identified only two weaknesses:(1) The authors often mixed the methods and the results, Which reduces the readability and fluidity of the manuscript. I would recommend the authors to re-structure the manuscript.(2) In some parts, the authors stated that they reconstructed the visual field of the pigeon, which is not true. They identified the foveal positions, but not the visual fields, which involve different sectors (binocular, monocular or blind). Similarly, they sometimes mix-up the area centralis and the fovea, which are two different visual adaptations.

Thank you for your positive feedback. We addressed these comments by restructuring the methods and result sections as suggested, and by checking the terminology and specific vocabulary used throughout the manuscript.

**Reviewer #1 (Recommendations For The Authors):**
First, I would like to say that I really enjoyed the manuscript. This is a great contribution to the field.

Thank you for the positive feedback, we highly appreciate it.

Then, I have some comments that I hope, would help the authors to improve the manuscript.Major comments :I would recommend the authors to restructure the methods and the results section. In many parts, the models used are presented in the results section, while this should be presented in the methods section.

Thank you for the suggestion, we now have ensured that the model descriptions are presented in the statistic section of the methods.

To me, the introduction is too long (more than 5 pages). It would be beneficial to reduce it considerably. Furthermore, in the introduction, it misses some information about the visual abilities of your species (visual acuity, visual field, temporal resolution, contrast sensitivity....).

We agree that the introduction was very long and reduced it by removing the “Methodological issues” as well as strongly reducing the “Experimental rationales” to a minimum. We also added the missing information on the visual abilities of the pigeons in the “Experimental rationales” section (see L135-150). Please note, however, that we refer to the temporal resolution of pigeon vision in the method section, to associate it with the information of the used monitor’s resolution.

Minor comments :Lines 37-39: This needs a reference.

A reference has been added (McFarland, 1977)

Lines 39-41: But see some papers published recently on Harris's hawks.

Thank you for the references, we added the citation as well as a few more papers (Kane et al., 2015; Kano et al., 2018; Miñano et al., 2023; Yorzinski & Platt, 2014).

Lines 41-43: This sentence needs a reference as well.

A reference has been added (Cresswell, 1994; M. H. R. Evans et al., 2018; Inglis & Lazarus, 1981)

Lines 56-103: In this paragraph, head down and head up also depends from the retinal map of the birds! Some birds have visual streak that allow them to see a potential threats while foraging. Please add more information about the importance of photoreceptors distribution.

Thank you for pointing out this issue. We rewrote the sentence L65-69 as follows to include the importance retinal structures.

“In several species, especially those with a broad visual field and specific retinal structures such as the visual streaks, individuals can simultaneously engage in foraging activities while remaining vigilant (Fernández-Juricic, 2012), likely using peripheral vision to detect approaching threats (Bednekoff & Lima, 2005; Cresswell et al., 2003; Kaby & Lind, 2003; Lima & Bednekoff, 1999).”

Lines 76-79: you wrote : ".... favor alternative hypotheses based on their findings". Which findings? You need to explain.

We rewrote this part as follows (L80-81).

“other studies found evidence for the risk dilution (Beauchamp & Ruxton, 2008) and the edge effect (Inglis & Lazarus, 1981) in their study systems.”

Lines 109-110: It would be good to have a representation of what is an area and a fovea, and how it is placed in the eye, what type of fovea exists and how it is related to visual field. Where does it project?

We now give a better description of the pigeon’s visual field in the experimental rationales section that we hope will help the reader understanding the key features of pigeon’s vision (see L135-150). Specifically, we now say in L137-138:

“they have one fovea centrally located in the retina of each eye, with an acuity of 12.6 c/deg (Hodos et al., 1985). Their fovea projects laterally at ~75° into the horizon in their visual field.”

Lines 109-113: You might need to see some new papers here about the fovea. See for instance Bringmann 2019.

Thank you for the suggestion, we now give a more precise definition of the fovea and refer to Bringmann’s paper for more details (L113-114):

“a pit-like area in the retina with high concentration of cone cells where visual acuity is highest, and is responsible for sharp, detailed, and color vision.”

Lines 113-120: Please explain how the visual field is related to fovea? Where is the fovea project in the visual fields?

Similarly to the question above, we now give a more precise description of the pigeon’s visual field (see L135-150).

Line 131-134: For a non-expert, you would need to explain what is micro, meso and macro scale?

These sentences have been removed when shortening the introduction and we are not referring to micro, meso and macro scales anymore.

Lines 134-136: Please explain in one sentence the technique here.

We now explain in one sentence how motion capture enables the tracking of head and body orientation (L130-132):

“Motion capture cameras track with high accuracy the 3D position of markers, which, when attached to the pigeon’s head and body, enables to reconstruct the rotations of the head and body in all directions.”

Line 140: You presented here for the first time the word "foveation". Has this term been used before? If so, please add a reference. If not, please explain what you mean by foveation precisely.

Thank you for noticing this lack. We are now providing the following definition “directing visual focus to the fovea to achieve the clearest vision” in the first place where we mention the term foveation (L149-150).

Lines 146-148: Please explain why this proves that it is appropriate to not record eyes movements, and is this true for every behaviours?

We acknowledge that some small eye movement might occur and reduce the accuracy of the method. This error is considered in the system using the +-10 degrees range around the foveas. The lines the reviewer referred to were removed when shortening the introduction, but we added an explanation in the paragraph describing pigeon vision to make it clearer (L147-150):

“Yet, it should be noted that their eye movement was not tracked in our system, although it is typically confined within a 5 degrees range (Wohlschläger et al., 1993). We thus considered this estimation error of the foveation (directing visual focus to the fovea to achieve the clearest vision) in our analysis, as a part of the error margin (see Methods).”

Lines 161-163: What is the frontal and binocular field for? You would need to explain the different fields of view and what they are supposed to be for.Furthermore, does the visual field of pigeon have been studied? If so, you would need to add more information about it.

This information is now given in the new paragraph describing the pigeon’s vision in the “Experimental rationales” section (see L135-150).

Figure 1: It is not clear here which panels correspond to a, b or c. Please use some boxes to clarify it.

Thank you for the comment, we now have made the figure’s sub-panels clearer.

Lines 193-194: You wrote "... such as foveas (also known as the area centralis). No, this is not the same.(1) In some species, you have two foveas, one placed centrally in the retina, one place temporally. So the fovea is not the area centralis.(2) Second, some species do have an area centralis but without a fovea.

Thank you for pointing out the inaccuracy. In this case, we were referring specifically to the pigeon’s fovea which is sometimes referred to as “area centralis”, but we now changed the sentence as follow to avoid any confusion (L174-175):

“The initial two hypotheses (Hypotheses 1 and 2) aim to examine whether foveation correlates with predator detection.”

Lines 192-212: I did not understand the logic of the hypotheses numbers? Why do you have 2.1 but not 3.1 for instance? And if you have two hypotheses for the within a global one (for instance, 2.1 and 2.2), what is the main hypothesis 2? You should explain more here because we get lost here and in the result section as well.

We recognize this section might have appeared confusing to the reader. In short, we had four main hypotheses: (1) the fovea is used to evaluate predator cues, (2) the latency to foveate is related to vigilance behaviors. These first 2 hypotheses aimed to determine if the latency to foveate on the predator cue could be related to the detection. (3) foveation is related to the escape response of the pigeons and (4) there is a collective influence in the escape response. We further divided some of the hypotheses into 2 sub-hypotheses whenever 2 different tests were used to answer the same question. We have modified this section to be clearer.

Lines 224-229: Where are the figures and statistics for these results?

These results are presented in Table S1. We apologize for forgetting to add this reference and have now added it (L211).

Lines 229-231: This should be in the method section.

This model explanation (as well as all other hereafter mentioned) have been moved to the method section as suggested.

Lines 248-252: This should be in the method section. Furthermore, you should better explain the model selection.

Please see earlier comment. Additionally, we are now better explaining how the model has been built.

Figure 2: It is not clear on the figure which letters correspond to which panels. Please improve the readability of the figure.

It was modified accordingly.

Lines 274-278: This should be in the method section.

Please see earlier comment.

Line 281: The "Fig.3" should be mentioned in the previous sentence.

It was modified accordingly.

Figure 3: Please explain why the latency to foveate had negative values in Fig.2 but not here, and not in Fig. 4 as well. This again highlights that we missed a number of information in the methods about the transformation of the data and the model selection.

The variable presented in Fig 2d is not the latency to foveate but the “Normalized frequency at which the object was observed within foveal regions” (hypothesis 1). It represents the amount of time the object was lying within one of the foveal regions of the individual (“how long the pigeons foveated on it”), further normalized to unit sum to make all objects comparable. This variable was indeed logit-transformed (hence the negative value) to improve residual fit in the model, but this information (as well as other transformations) are always clearly stated on the axis caption of the graphs. Additionally, we now have improved the statistical analysis section to make the model used for each hypothesis testing clearer. But please let us know if you have suggestions for a further improvement in terms of presentation.

Lines 297-301: This should be in the method section.

Please see earlier comment.

Lines 301-305: Fig. 3 b and c only referred to the two first factors. Please add more figures for the other factors. This could be in supp. Mat.

We added the 3 graphs for the proportion of time foveating on the monitor, the saccade rate and the proportion of time foveating on conspecifics in the supplementary (Fig S6).

Lines 306-309: This should be in methods, and you should have explained in methods how you performed your model selection.....

We prefer leaving this paragraph in the result section, as it was intended to give the reader extra information on the predictive power of the different variables (by comparing the effectiveness of the models including one variable at a time, all the rest being equal) and not on the model selection *per se.* However, we now explain our goal better in the statistics section regarding this analysis (L635-636):

“We further tested the relative predictive power of the different test variables by comparing the resulting models’ efficiency using AIC scores.”

Lines 317-319: This should be in the method section.

Please see earlier comment.

Lines 320-322: This should be in the method section.

Please see earlier comment.

Lines 332-334: This should be in the method section.

Please see earlier comment.

Lines 334-336: Then, if this is not significant, you cannot say that.

Thank you for noticing the inaccuracy, we have now rephrased it as (L298-299):

“Earlier foveation of the first pigeon was not significantly related to an earlier escape responses among the other flock members, although there was a trend (χ2(1) = 3.66, p = 0.0559).”

Line 336: Please explain why you did different models. We missed a lot of information in the method about your strategy for statistics.?

We have now added a lot more information on the models in the statistics section, according to this comment as well as the previous ones. We hope the explanations of the analyses are now clearer to the reader.

Lines 339-349: This should be in the method section.

Please see earlier comment.

Results section: As you may have understood, there are too many sentence that should be moved into the method section. Futhermore, I would recommend to modify the headdings so that they are more biologically speaking. Similarly to what you have done in the discussion section.

Thank you for the comments. We agree with most of them, and have modified the manuscript accordingly. Additionally, we now use the same headings in the results section as the ones used in the discussion to make the text easier to follow.

Lines 500-501: What were the body weight of the pigeon? At which weight of their full weight they were?

This information is now added (492 ± 41g; mean ± SD). We did not control the amount of food during our experiments and only ensured 24h without food by feeding the pigeons after the experiment was completed. This information was added as follows (L454-456):

“On experimental days, they were fed only after the experiments was completed; this ensures 24-hour no feeding at the time of the experiment, although we did not control the amount of the food over the course of the experimental periods.”

Line 522-523: Those screens are very good for pigeons.

Thank you for the positive comment, we indeed tried to match bird vision as close as possible.

Lines 527-528: At which frequency was produced the moving stimulus? Your screen can display up to 144Hz, which is very good. But can your laptop do it? If not, it is important to mention it as pigeons may have a temporal resolution of vision up to 149Hz.

Our laptop indeed supports 144Hz display. In addition, we now mention the temporal resolution of pigeon vision (L480-482).

“We specifically chose a monitor with high temporal resolution to match the pigeon’s Critical Flicker Fusion Frequency (threshold at which a flickering light is perceived by the eye as steady) that reaches up to 143Hz (Dodt & Wirth, 1954).”

Lines 555-572: Did you use a control shape in your experiment? Indeed, they may escape because of a moving pattern but not a predator shape?

We did not use a control shape, as the aim of the experiment was not to directly test the effect of the shape itself. We designed the predator cue to resemble an approaching predator to ensure a response from the pigeons, but it might be that other shapes would have worked as well.

Lines 588-589: Please explain why the coordinate system of the pigeon's head is considered as the visual field?From what I have understood, you did not reconstruct the visual fields, but only the position of the fovea. This should be noted like this as visual field involves more than a sphere around the head (binocular and monocular sectors, blind sectors, vertical extension....).

Thank you for noticing the inaccuracy, we indeed did not consider other sectors of the visual field and therefore rephrased it as (L551): “the location of the objects and conspecifics from the pigeon’s perspective”.

Lines 601-604: How much does it represent?

As this was estimated by visual inspection, we do not have the exact percentage of data loss that was caused by grooming. However, because of the number of cameras in the SMART BARN motion capture system, it is reliable in detecting markers inside the space in “ideal” conditions (without occlusion). For example, a similar set-up found marker track loss of only <1% using a model bird (Itahara & Kano 2022)

Itahara, A., & Kano, F. (2022). “Corvid Tracking Studio”: A custom-built motion capture system to track head movements of corvids. Japanese Journal of Animal Psychology, 72(1), 1–16. https://doi.org/10.2502/janip.72.1.1

Lines 610-612: You would need to cite Wood 1917 and Hodos et al. 1991 who described the presence of a fovea in this species.

We added both citations to the manuscript.

Line 611: Again, the fovea is not egal to area centralis.

Thank you, we changed it as well.

Lines 625-626: you wrote "... in a few instances....". Please explain more. How many? What proportion?

This happened in 9 observations out of 120. We now specify it in the text as well (L587-589):

“in a few instances (9 out of 120 observations), pigeons foveated on the model predator after the looming stimulus had disappeared, but these cases were excluded from our analysis.”

Lines 640-653: We missed a lot of information in the section "statistical analysis". If you moved most of the sentence from the results that describe the methods in the method section, that would be much better. Furthermore, you would need to explain more what statistics you used, which model selection, what type of data transformation....

We agree this section lacked information, and we moved the information from the result to the statistics section.

Supplmentary materials: boxplots from Fig. S1 and S2 are too small and impossible to read. Please improve the readability.

We now have enlarged these plots to make them more readable.